# Geomapping Vitamin D Status in a Large City and Surrounding Population—Exploring the Impact of Location and Demographics

**DOI:** 10.3390/nu12092663

**Published:** 2020-08-31

**Authors:** Helena Scully, Eamon Laird, Martin Healy, James Bernard Walsh, Vivion Crowley, Kevin McCarroll

**Affiliations:** 1Mercer’s Institute for Research on Ageing, St. James’s Hospital, Dublin 8, Ireland; jbwalsh@tcd.ie (J.B.W.); kmccarroll@stjames.ie (K.M.); 2School of Medicine, Trinity College Dublin, Dublin 2, Ireland; Lairdea@tcd.ie; 3Department of Biochemistry, St James’s Hospital, Dublin 8, Ireland; mhealy@stjames.ie (M.H.); vcrowley@stjames.ie (V.C.)

**Keywords:** vitamin D, 25(OH)D, vitamin D deficiency, geomapping, Ireland, Europe

## Abstract

Vitamin D status was assessed in a large urban area to compare differences in deficiency and to geomap the results. In total, 36,466 participants from 28 geographical areas were identified in this cross-sectional, retrospective analysis of general practitioner (GP)-requested 25(OH)D tests at St James’s Hospital, Dublin between 2014 and 2018. The population were community-dwelling adults, median age 50.7 (18–109 years) with 15% of participants deficient (<30 nmol/L), rising to 23% in the winter. Deficiency was greatest in younger (18–39 years) and oldest (80+ years) adults, and in males versus females (18% vs. 11%, *p* < 0.001). Season was the biggest predictor of deficiency (OR 4.44, winter versus summer, *p* < 0.001), followed by location (west Dublin OR 2.17, north Dublin 1.54, south Dublin 1.42 versus rest of Ireland, *p* < 0.001) where several urban areas with an increased prevalence of deficiency were identified. There was no improvement in 25(OH)D over the 5-year period despite increased levels of testing. One in four adults were vitamin D deficient in the winter, with significant variations across locations and demographics. Overall this study identifies key groups at risk of 25(OH)D deficiency and insufficiency, thus providing important public health information for the targeting of interventions to optimise 25(OH)D. Mandatory fortification may be necessary to address this widespread inadequacy.

## 1. Introduction

Vitamin D has become the focus of increased interest globally, with the number of web searches rising year on year, peaking in the winter and now eclipsing that of Vitamin C [1]. Vitamin D has an established role in maintaining normal bone health, being required for the adequate absorption of calcium and phosphate from the gut and thereby mineralisation of the skeleton. Deficiency causes rickets in children and osteomalacia in adults and it can also exacerbate or contribute to the development of osteoporosis. Furthermore, vitamin D deficiency can lead to muscle weakness and may increase the risk of falls and fractures [2]. More recently, research has demonstrated associations with chronic conditions such as diabetes [3], inflammation [4], cardiovascular disease [5], depression [6] and cancer [7].

Vitamin D is unique as it is the only vitamin that can be synthesised endogenously via the action of ultraviolet-B (UVB) light on the skin. In fact, the majority (90%) of our vitamin D is derived in this way, making it a highly variable source. Geographical latitude, time of year, cloud cover, sunscreen use, skin pigment, obesity, religious dress and age can all affect UVB vitamin D synthesis [2,8]. In locations greater than 30° north or south latitude, a ‘vitamin D winter’ exists between October and March when little or no vitamin D can be produced due to limited UVB penetration [9]. During this period, we rely on vitamin D through diet alone though in countries such as Ireland, where there is no mandatory vitamin D fortification of foods, it is difficult to reach the recommended daily intake (10 µg/day) as sources in the diet (such as oily fish) are limited or are often not consumed [10]. As a result, a significant proportion of the population is at risk of deficiency (25-hydroxyvitamin D {25(OH)D < 30 nmol/L}) [11]. In particular, those most at risk include indoor or night-shift workers [12], housebound or noncommunity dwellers [13] or the elderly with reduced capacity for cutaneous synthesis [14].

It is difficult to compare studies of vitamin D status as they involve different populations and thresholds for defining deficiency. In Ireland, the National Adult Nutrition Survey (*n* = 1132) found that 21% of 18–84-year-olds were deficient (<30 nmol/L), although this included a small number of older adults [10]. The only other nationally representative Irish study was The Irish Longitudinal Study on Ageing (TILDA) (*n* = 5356), which found that 13.1% were deficient (<30 nmol/L), rising to 23% in the winter [14]. One study of Irish rural and urban dwellers (*n* = 17,590) identified that 15.9% of adults were deficient (<25 nmol/L) but included only those in the west of the country [15]. The Trinity Ulster Department of Agriculture (TUDA) Study of older Irish adults (>60 years, *n* = 4444) reported that between 13.8% to 43.6% were deficient (<30 nmol/L), though had participants from disease defined cohorts [16]. However, it is not only older adults who are at risk. For example, in a study of adolescents across 9 EU countries, 15% were deficient (<27.5 nmol/L) and 27% insufficient (27.5–49.9 nmol/L) [17].

Environment is an important determinant of health and when combined with other data can be used to create geomaps. To date, few studies have applied this technique when exploring vitamin D status. Of note, geomapping of a large urban area in Calgary, Canada identified population clusters where education and immigration status were the strongest predictors of 25(OH)D [18]. Moreover, vitamin D status has also been geomapped in a population of free-living adults in Dublin, Ireland (53 N°). This study found that 15.2% were deficient (<30 nmol/L) in winter, improving to 10.8% in the summer, but with significant variation by postal code area [19]. However, the vitamin D measurement was limited to a one-year period.

The current study aims to investigate the vitamin D status of community-dwelling Irish adults over a 5-year time period (across a broad age range) living in Dublin and surrounding areas who had their vitamin D tested in primary care by request of their general practitioner (GP). A key objective is to explore the effects of gender, age, season and geographical area on vitamin D status. Finally, in what is the largest study of its type in Europe, we aim to create a geomap that visually depicts the prevalence of vitamin D deficiency by location.

## 2. Materials and Methods

### 2.1. Data Collection

St James’s hospital (SJH) is the largest academic teaching hospital in the Republic of Ireland serving a population of approximately 350,000 people. It is located in Dublin city, on the east coast of Ireland (53.35° North latitude) and receives the majority of referrals from Dublin city and the greater Dublin area. A search was completed for vitamin D requests from primary care GPs at the SJH biochemistry department via its information system (iSOFT Telepath^®^). Samples requested between 2014 and 2018 (inclusively) were selected for a retrospective cross-sectional analysis. The exclusion criteria were: aged <18 years, missing or incomplete demographic data, noncommunity dwelling address (e.g., nursing home or hospital) or address outside of the Republic of Ireland. Repeat vitamin D results were excluded to avoid pseudo-replication.

Dublin area postal codes were used to record participants residence. Dublin areas are represented by postal codes (D1 to D24) with odd numbers for locations in north Dublin and even numbers in south Dublin. Some locations have no postal code but a specific name or are known only as being in north or south. County Dublin was also categorised into three main areas: North Dublin (D1, D3, D5, D7, D9, D11, D13/17, North County Dublin); south Dublin (D2, D4, D6/6W, D8, D10, D12, D14/16, D18, south County Dublin); west Dublin (D15, D20, D22, D24, Lucan). County Kildare was split into north (including the towns of Leixlip, Maynooth, Celbridge and Kilcock) and rest of Kildare. Residents in Counties Meath and Wicklow were also designated separately. Participants living in the province of Leinster (but not in Dublin or adjacent counties of Meath, Kildare and Wicklow) were classified as ‘rest of Leinster’. Those residing outside of the above locations were categorised as ‘rest of Ireland’.

### 2.2. Ethics

The joint research ethics committee at St James’s Hospital/Tallaght University Hospital (SJH/TUH) granted ethical approval for this study (Ref: 5475) which was conducted according to the guidelines laid down in the Declaration of Helsinki 1964.

### 2.3. Serum 25(OH)D Measurement

The nutritional marker of vitamin D status, serum concentration of 25(OH)D (total 25(OH)D2 and 25(OH)D3), was quantified by liquid chromatography-tandem mass spectrometry (LC-MS/MS) (API 4000; AB SCIEX) using a fully validated method (Chromsystems Instruments and Chemicals GmbH, Gräfelfing, Germany MassChrom 25-OH-Vitamin D3/D2) at the Biochemistry Department of SJH, which is fully accredited to ISO 15189:2012 standard. Quality is monitored by assay of internal quality controls, participation in the Vitamin D External Quality Assessment Scheme (DEQAS) and the utilisation of the National Institute of Standards and Technology (NIST) 972 25(OH)D standard reference material (SRM 972) to determine accuracy. The limit of quantification was 9 nmol/L, with values below this assigned as 9 nmol/L. The respective inter-and intra- assay coefficients of variation are 5.7% and 4.5%.

There is a lack of agreement as to what constitutes vitamin D deficiency or suboptimal vitamin D status [20,21]. The Institute of Medicine (IOM) defines risk of deficiency at as <30 nmol, 30–49.9 nmol/L as being at risk of inadequacy and replete status at ≥50 nmol/L [11] while others define deficiency as <25 nmol/L [2]. A 25(OH)D level >125 nmol/L may be harmful to health [11,22]. In this study, we defined deficiency as <30 nmol/L, insufficiency as 30.0–49.9 nmol/L and sufficiency as ≥50 nmol/L, as used elsewhere [11,19]. Participants with 25(OH)D level >125 nmol/L were also identified. Seasons were defined as winter (December, January, February), Spring (March, April, May), Summer (June, July, August) and Autumn (September, October, November).

### 2.4. Statistics

Statistical analysis was carried out using SPSS (Version 24, IBM Corp., Armonk, NY, USA.) Data were checked for normality by the Kilmogorov-Smirnov test and Q-Q plot and transformed where necessary. Data reported in tables and maps are expressed as geometric mean with standard deviation (SD). Independent sample *t*-tests, one-way ANOVA for continuous and Chi-square for categorical variables were performed as appropriate to assess statistical significance (*p* < 0.05). Postal districts with smaller populations (*n* < 100) but adjacent to each other and with similar demographics were combined for the analysis (D6/D6W, D13/17 and D14/16). Multinomial regression was used to explore the determinants of 25(OH)D status including age, gender, season of sampling and geographical area. The areas of ‘rest of Leinster’ and ‘rest of Ireland’ were combined to form ‘outside Dublin’ as the reference area.

### 2.5. Geomapping of Participants

A colour-coded map was created depicting the prevalence of deficiency for the areas sampled in the seasons of summer and winter, with categorisation based on the prevalence rates of 25(OH)D<30 nmol/L as follows: <10%, 11–20%, 21–29% and >30%. Areas with smaller numbers (*n* < 120) that were likely to be nonrepresentative were excluded from our map.

An ANOVA analysis was used to identify mean differences in 25(OH)D between areas in the winter and summer (see Appendix B), and cluster maps were created. These depict differences between 25(OH)D across several areas, categorised with equal distribution based on median values for winter (<10 nmol/L, 10–20 nmol/L, >20 nmol/L) and summer (<10 nmol/L, >10 nmol/L).

## 3. Results

### 3.1. Demographics

There were 51,651 serum 25(OH)D results reported to primary care GPs between 2014 and 2018, of which 36,466 (70.6%) met the inclusion criteria (Figure 1). The population and area demographics are shown in the respective Table 1 and Table 2. There was a relatively even distribution of samples across all seasons. The median age was 50.7 years, ranging from 18 to 109 years. We sampled 28 areas, with requests from County Dublin and Kildare comprising the majority (97%). Approximately 15 areas had 25(OH)D results for at least 500 people, Dublin 13/17 being the smallest (*n* = 107) and north Kildare the largest (*n* = 5734). Those in Dublin 1 were the youngest (35.9 ± 11.6 years) with 88% aged under 50 years, while those in Dublin 20 were the oldest (57.8 ± 17.5 years). The majority of requests (72%) were for females. This varied by location, ranging from 64% in Dublin 20 to 80% in Dublin 3.

The 25(OH)D geometric mean was lowest in winter (46.8 ± 30.3 nmol/L) and highest in autumn (63.0 ± 29.4 nmol/L) versus spring (48.9 ± 30.4 nmol/L) or summer (58.6 ± 30.5 nmol/L). Females had higher 25(OH)D versus males (55.1 ± 31.3 nmol/L vs. 49.7 ± 29.0 nmol/L, *p* < 0.001). 

### 3.2. 25(OH)D Status by Year, Age and Gender

The 25(OH)D status over the five-year period is shown in Table 3, with the proportion in each category (deficient, insufficient and >125 nmol/L) split by age and gender. Overall, there was a 58% increase in vitamin D testing from 2014 to 2018. The proportion who were deficient, insufficient and who had 25(OH)D >125 nmol/L was relatively stable over time at 15%, 23% and 3% respectively. When dichotomised by age, we found that those <50 years had a higher level of deficiency versus those ≥50 years (18% vs. 11%, *p* < 0.001).

Those aged ≥50 years were more likely to have a 25(OH)D > 125 nmol/L (4% vs. 2%, *p* < 0.001) as were females (3% vs. 2%, *p* < 0.001) and those sampled in the summer (*p* < 0.001). Overall, the prevalence of deficiency was greater in males versus females (17% vs. 14%, *p* < 0.001) as was insufficiency (27% vs. 22%, *p* < 0.001).

Figure 2 illustrates 25(OH)D status in various age categories. Those who were youngest (18–39 years) had the highest prevalence of deficiency (21%) and insufficiency (26%). This prevalence was only matched by those in the very oldest age group (>90+ years). In fact, there was a ‘U’ shaped relationship, with the best vitamin D status in those aged 60–69, and then progressively declining when moving towards both the younger and older ends of the age spectrum.

### 3.3. (OH)D Status by Area—Effect of Season

The prevalence of deficiency and insufficiency is shown for each area and categorised by season (see Table 4). Deficiency was greatest in winter at 23%, with a further 26% insufficient. In contrast, deficiency was lowest in summer at 8%, with an additional 16% insufficient. The locations with the lowest 25(OH)D were; Dublin 1 (45.5 ± 31.4 nmol/L), Dublin 11 (46.1 ± 27.3 nmol/L), Lucan (46.5 ± 30.8 nmol/L) and Dublin 8 (49.1 ± 31.9 nmol/L). The locations with the highest 25(OH)D were Dublin 14/16 (59.9 ± 30.8 nmol/L), Dublin 4 (58.8 ± 33.0 nmol/L), Dublin 6/6W (58.6 ± 31.1 nmol/L).

In winter, most locations had a prevalence of deficiency of 20% or more. In particular, in this season the areas (D1, D2, D7, D10, D11, Lucan, North County Dublin) and County Wicklow had more than 30% who were deficient. A select number of locations (D4, D6, D6W, D14/D16, North Kildare) had a deficiency of 11–20% in winter while no areas had a prevalence of less than 10%. In summer, some areas had no deficiency (County Wicklow/Dublin 9) while in others such as Dublin 5 the prevalence was as high as 16%.

A geomap gives a visual representation of the prevalence of deficiency by location in the summer and winter (Figure 3 and Figure 4). This highlights a widespread deficiency in Dublin (20- >30%) and surrounding counties in the winter (Figure 3). However, the opposite is true in the summer where most areas have a prevalence of deficiency less than 10% (Figure 4).

### 3.4. 25(OH)D Status by Area—Age and Gender

There were significant differences in 25(OH)D status when dichotomised by age across the areas (Table 5). In every area, deficiency was more prevalent in those who were younger (<50 years), with this being more marked in some locations. For example, in Lucan, Dublin 1, Dublin 8 and Dublin 22, more than 20% of those aged <50 were deficient. Similarly, in the same areas and also in Dublin 11 more than 50% of this age group had a level below 50 nmol/L. Conversely, in those aged ≥50 only two areas, Dublin 13/17 and Dublin 2, had a level of deficiency and insufficiency above 20%. In just over half of the locations, males were more likely to be deficient, in keeping with the overall study findings. The prevalence of 25(OH)D >125 nmol/L ranged from 1–7% in females, to 0–4% in males and was greatest (7%) in women living in Dublin 3. 

### 3.5. Determinants of Vitamin D Deficiency and Sufficiency

The independent effects of gender, season and location on 25(OH)D status are outlined in Table 6. Season was the strongest predictor of deficiency followed by geographical area and then gender. Those sampled in the winter versus the summer were over four-times more likely to be deficient (OR 4.43, *p* < 0.001). In terms of location, those living in north Dublin versus outside Dublin were more likely to be deficient (OR 1.54, *p* < 0.001) and insufficient (OR 1.089, *p* < 0.001) while those in west Dublin were more than twice as likely to be deficient (OR 2.17, *p* < 0.001). We also identified that females were 32% less likely to be deficient (OR 0.68, *p* < 0.001).

### 3.6. 25(OH)D Status versus Season, Dichotomised by Age and Gender

The seasonality of 25(OH)D is shown in Figure 5 where the geometric mean for each season over 5 years is illustrated. Seasonality was similar regardless of age or gender. However, 25(OH)D is consistently higher for those over 50 (mean difference + 10.9 nmol/L) and females (mean difference + 5.5 nmol/L) across all seasons (Figure 6 and Figure 7).

### 3.7. Cluster Analysis of Differences in 25(OH)D between Areas (See Appendix A)

Cluster maps of postcodes with significant mean differences in 25(OH)D in the winter (Appendix A) and summer (Appendix A) were created. There were more areas with significant mean differences in the winter (Appendix A and S2). In winter, higher mean 25(OH)D were identified in the areas (D4, D6/6W, D14/16) with significantly lower levels in central (D1, D2, D8), west (D10, D22, Lucan) and north Dublin (North Co. Dublin, D11). In summer, the areas D4, D6/6W and D14/16 had higher status versus Dublin city and west Dublin. Finally, North Kildare had a higher 25(OH)D in the summer compared with the areas of Lucan and Dublin 8 and this also remained the case versus Lucan in the winter.

## 4. Discussion

To our knowledge this is the largest geomapping study of vitamin D status in Europe. We observed that nearly one in six (15%) of a GP primary care tested-population in Dublin and surrounding areas were vitamin D deficient, rising to one in four (23%) in the winter. Furthermore, an additional 26% were insufficient (30–49 nmol/L), with nearly half of those tested having 25(OH)D levels <50 nmol/L in the winter. We also identified major differences in the prevalence of deficiency between Dublin postal code areas, despite being in close proximity to each other, as well as in other counties. This is concerning as it suggests that a significant proportion of Dublin and surrounding area population have inadequate 25(OH)D status.

### 4.1. Vitamin D Status by Gender

We observed that overall, females had higher vitamin D serum levels across all age groups, similar to other studies [14,15,23]. In fact, females were about a third (32%) less likely to be deficient. Some research conflicts with this [24,25] although a meta-analysis of 394 studies (*n* = 33,266) discovered higher mean 25(OH)D in women, with levels comparable to our study [26]. In about half of locations the difference in 25(OH)D between genders was not statistically significant, but the analysis may have been underpowered due to smaller sample sizes in some areas. In addition, other factors related to gender which we were not able to adjust for might account for this.

The majority of vitamin D requests (72%) were for females, a finding which has been reported elsewhere [15,19]. Females may be more likely to attend their GP and partake in positive health behaviours [27]. They are also more likely to take a dietary supplement [28]. For example, in one study, females (particularly those aged over 50) were more than twice as likely to routinely take a vitamin supplement [29,30]. Furthermore, there is a greater awareness of the importance of bone health in women, where osteoporosis is more prevalent and emphasis is placed on prevention and treatment [31].

### 4.2. Vitamin D Status by Age

Those who were younger (<50 years) had lower vitamin D status. This did not change over the 5-year period, with about 1 in 5 of those aged < 50 being deficient compared to 1 in 10 over 50. Surprisingly, those aged between 18 and 39 years had the same prevalence of deficiency (21%) as the oldest age group (+90 years). This nadir in 25(OH)D in young adults was also found in the similarly located northern latitude (51° N) city of Calgary, Alberta [18]. A ‘U’-shaped distribution of vitamin D deficiency in the youngest and oldest adults has likewise been described in the west of Ireland [15], Romania [32], and São Paulo, Brazil [33].

One reason why younger adults may be deficient is that they may spend more time indoors e.g., in their working environment. A study by Sowah et al., 2017 found that shift workers, healthcare and indoor workers have a higher risk of deficiency due to a lack of opportunity for sunshine exposure [12]. Another factor in younger adults may be the difference in dietary intake of vitamin D. In a recent Irish dietary survey, it was estimated that 19.2% of the population are now vegan, vegetarian or seeking to reduce dietary animal products with the majority of those being younger (aged 18–34) [34]. This may be a cause for concern as meat is the largest contributor of vitamin D in adults under 65, accounting for a third of total dietary intake [35]. Moreover, when compared to those over 65, the proportion of adults who did not meet the recommended dietary intake at that time (5 µg/day) was greater in the age group (18–64) for both males (72% vs. 59%) and females (78% vs. 58%) [35].

Ensuring adequate vitamin D status in younger adults is important, as peak bone mass is acquired in the early-to-late twenties [36]. Suboptimal 25(OH)D earlier in life and over prolonged periods might also contribute to other adverse health outcomes as consistent with the theory of ‘long latency deficiency disease’ [37]. For example, vitamin D deficiency has been associated with upregulation of inflammatory markers, endothelial dysfunction and chronic, low-grade inflammation that may increase the risk of cardiovascular disease as well as mortality from cancer and other causes [38,39].

Our findings demonstrate that 25(OH)D peaked in the decade 60–69 years in which most people retire and may reflect more time spent outdoors [40]. Conversely, 25(OH)D levels declined from the age of 70 onwards. This may largely relate to increasing frailty and less sunshine exposure. However, reduction in the capacity of the skin to synthesize vitamin D by up to 75% with age may also help to explain this [41]. In addition, sequestration of vitamin D within increased body fat with ageing as well as reduced dietary intake may contribute [42]. Furthermore, poor compliance with vitamin D supplements in older adults, especially in those where there is polypharmacy may also be a factor [43].

### 4.3. Vitamin D Status by Season and Location

A geomap depicting 25(OH)D status in Dublin and the surrounding areas illustrates major variations in the prevalence of deficiency by location. This varied greatly especially in the winter, where it ranged from 18% in areas like Dublin 14/16 and North Co. Kildare to up to >37% in Dublin 1 and Dublin 11. Findings were similar to a previous geomapping study of Vitamin D status in Dublin, although that study focused on a substantially smaller population, did not include as many areas and only covered a one year period [19].

In the winter, we found clusters of areas south Dublin (D4, D6/6W, D14/16) with greater serum 25(OH)D compared to west Dublin, north Dublin and more central Dublin city areas. In fact, these particular locations in south Dublin are some of the most affluent areas in the county and nationally, as determined by HP (Haase-Pratschke) Pobal Score 2016, a deprivation index of demographic profile, social class composition and labour market situation [44]. Socioeconomic status and 25(OH)D have been closely linked, with those in typically disadvantaged areas having an increased risk of deficiency [18,45]. For example, in the TILDA study of Irish adults, those with lower asset wealth were 1.5-times more likely to be deficient. This may be due to factors linked to lower socioeconomic status including reduced dietary vitamin D intake, less sunny holiday travel and possibly higher rates of obesity and smoking [14,46].

In the summer, most areas had a prevalence of deficiency of less than 10% except locations such as north and west Dublin including Lucan. This may be in part accounted for by a greater proportion of individuals of Asian and Black ethnicity living here [47]. Indeed, those with darker or more pigmented skin have a greater risk of deficiency at locations in northern latitudes [26]. However, overall season had a large impact on 25(OH)D in keeping with those in the winter being over four-times more likely to be deficient versus in the summer [22,48]. Importantly, the extent of the differences in deficiency between areas appeared to be attenuated in the summer, highlighting the importance of sun exposure.

Those living in the areas of north and west Dublin were also more likely to be deficient compared to those living outside Dublin. While we did not define rural or urban areas, locations outside Dublin are less urbanised and some are rural. These findings are in contrast to a study by Griffin et al., 2019, which found that Irish urban dwellers had higher 25(OH)D and lower rates of deficiency [15]. However, these differences might be accounted for by local variations in socioeconomic status and ethnicity.

There was no improvement in vitamin D status over a five-year period despite increased testing and greater awareness [1]. This suggests that a large proportion of the population have inadequate 25(OH)D but have not yet been identified. As the list of potential comorbidities related to vitamin D expands beyond bone and muscle health, it is important that no subgroup of the population is left vulnerable to deficiency. For example, vitamin D may help to maintain immune function, reduce the risk of respiratory infections and downregulate inflammatory cytokines, suggesting that deficiency could have a negative impact on Covid-19 outcomes [40,49]. Furthermore, vitamin D is a known regulator of cardiovascular and renal function mediated via the interaction of Vitamin D receptors within the renin-angiotensin-aldosterone system (RAAS), highlighting the multisystemic effects of vitamin D deficiency [50].

We did find that a small (1–7%) but not insignificant proportion of our study population had 25(OH)D levels >125 nmol/L, particularly females over 50 who may be more likely to take prescribed or over-the-counter vitamin D. While a similar prevalence (>125 nmol/L) has been identified elsewhere [22], it is important that any future Irish guidance on vitamin D intake takes into account the potential for inadvertently increasing the risk of hypervitaminosis D.

There are no national Irish guidelines regarding vitamin D testing, and recommendations for supplementation are limited to infants. This study highlights the presence of vulnerable groups with vitamin D deficiency, including males and young adults, living in poorer socioeconomic areas. Furthermore, our geomap and cluster analysis highlights locations where the prevalence of deficiency is significantly higher. These findings provide valuable information for GPs and public health bodies in developing strategies for targeted interventions to optimise vitamin D status.

### 4.4. Strengths of Study

The primary strength of this study was its large population size and the novel use of a geomap to gain a visual representation of vitamin D status in a large city and its surrounding areas. In fact, to our knowledge, this is the largest study in Europe to geomap vitamin D and includes data collected over a 5-year period. We used the gold standard for vitamin D assessment; LC-MS/MS, NIST internal standard and are DEQAS accredited to ensure accuracy.

### 4.5. Limitations of Study

Our study was based on GP vitamin D requests and this limits the generalisability of the findings to a wider population. In particular, there is likely to be selection bias in testing with study participants having medical conditions or other factors putting them at risk of deficiency. However, unlike other studies we minimised the potential for this by not including vitamin D samples received from outpatient or acute hospital services and also excluded those from institutionalised adults.

We were not able to adjust for several factors that influence 25(OH)D including biophysical (BMI, skin type, medical conditions e.g., malabsorption syndromes) and lifestyle (supplementation, smoking, sun exposure, alcohol intake, diet, education). Finally, our study was cross-sectional and therefore we could not make any direct inferences as to the causality of factors influencing vitamin D status.

## 5. Conclusions

This study shows that nearly one in four of the GP-tested population in Dublin and surrounding areas are vitamin D deficient in the winter, during which time up to 50% have 25(OH)D less than 50 nmol/L. Moreover, those living in poorer socioeconomic areas were more likely to be deficient, as are males, younger (18–39 years) and older adults (80+ years). This study identifies key groups at risk of vitamin D deficiency and provides important public health information for the targeting of interventions to optimise vitamin D status. Our data indicate that inadequate vitamin D status is not just limited to older adults, but it is widespread across many population groups. This highlights the importance for recommendations for vitamin D intake to include the entire population and/or mandatory vitamin D food fortification.

## Figures and Tables

**Figure 1 nutrients-12-02663-f001:**
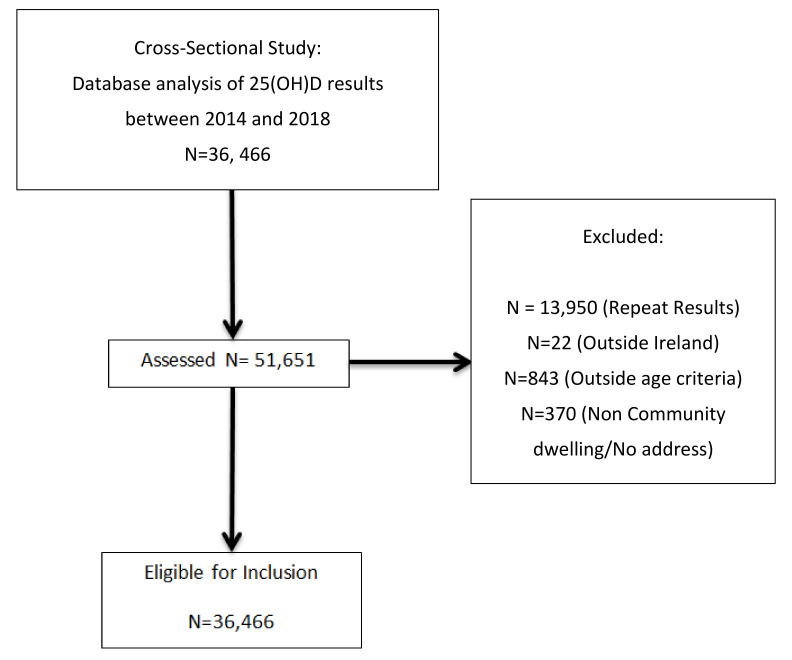
Recruitment Flow Diagram.

**Figure 2 nutrients-12-02663-f002:**
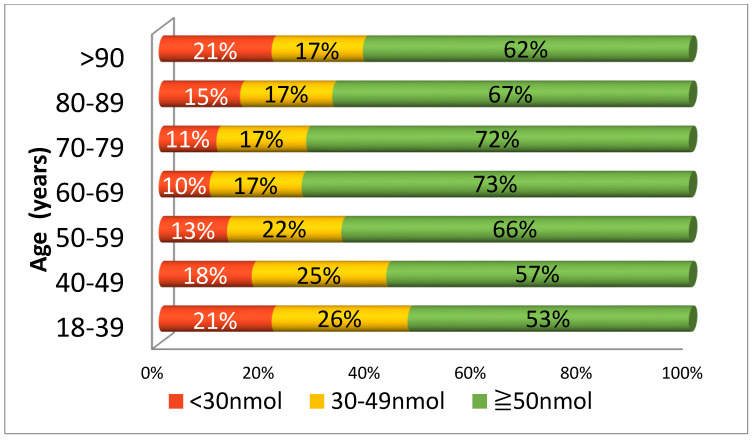
25(OH)D status in different age categories.

**Figure 3 nutrients-12-02663-f003:**
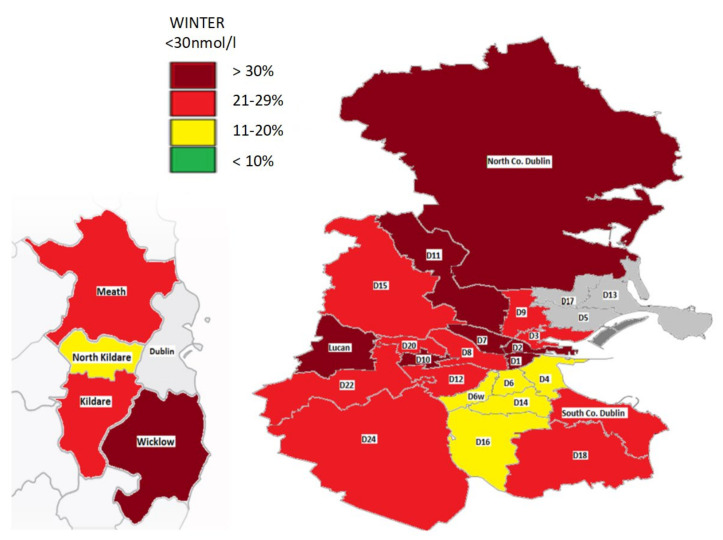
Geomap of 25(OH)D status in Dublin and surrounding counties in winter. Areas with insufficient numbers (<120) for analysis are shown in grey.

**Figure 4 nutrients-12-02663-f004:**
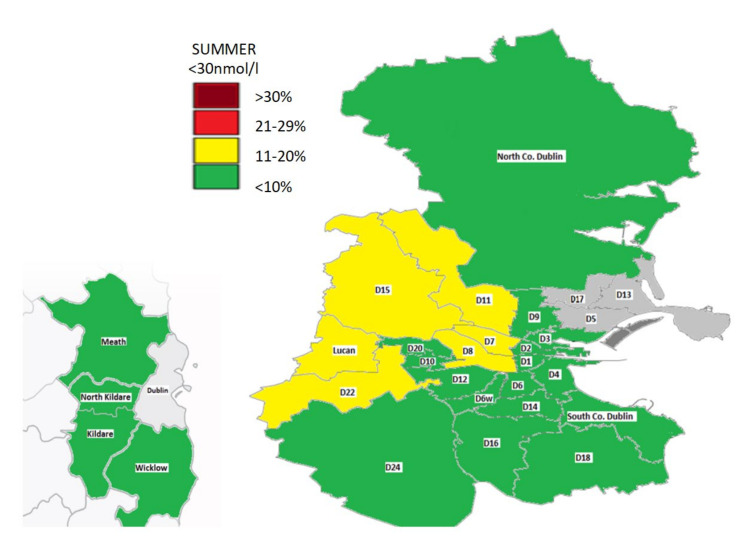
Geomap of 25(OH)D status in Dublin and surrounding counties in summer. Areas with insufficient numbers (<120) for analysis are shown in grey.

**Figure 5 nutrients-12-02663-f005:**
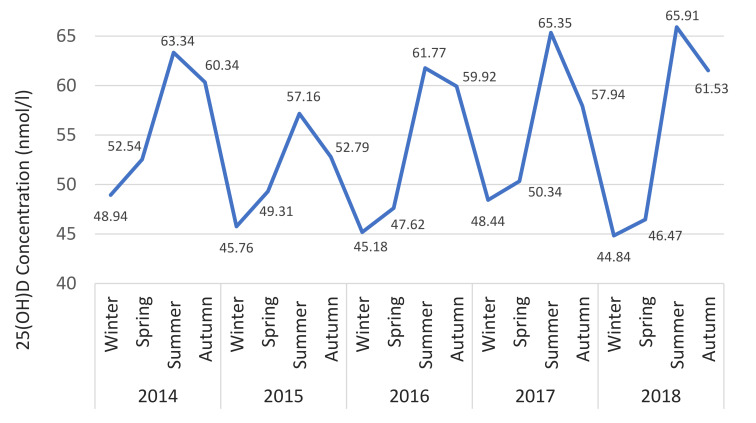
25(OH)D status versus season.

**Figure 6 nutrients-12-02663-f006:**
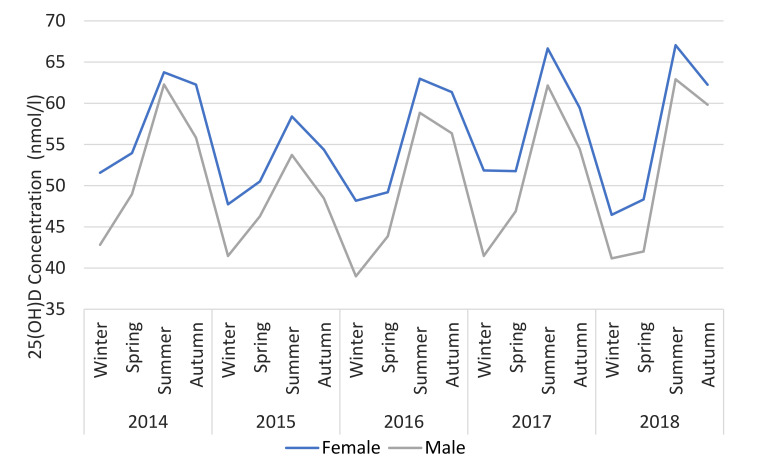
25(OH)D status versus season, dichotomised by gender.

**Figure 7 nutrients-12-02663-f007:**
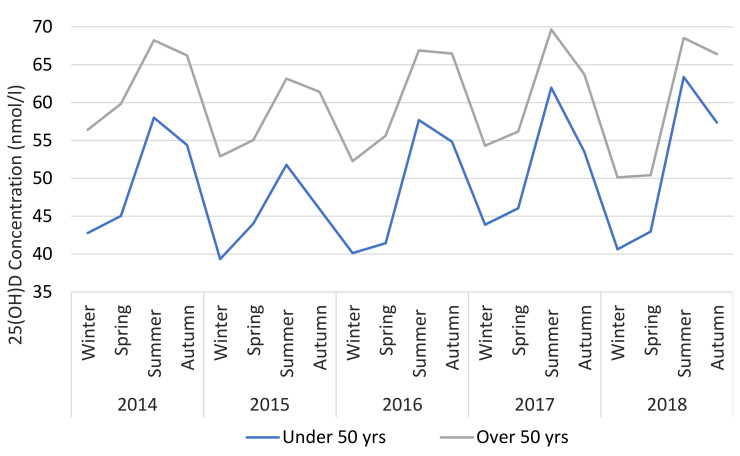
25(OH)D status versus season, dichotomised by age.

**Table 1 nutrients-12-02663-t001:** Demographics.

Demographics		N	%	GM Mean (SD)
Age Category (years)	18–39	11,319	31	47.8 (30.8)
	40–49	6977	19	50.3 (28.5)
	50–59	6328	17	56.5 (29.9)
	60–69	5753	16	62.1 (30.0)
	70–79	3865	11	61.7 (31.8)
	80–89	2003	5	58.1 (33.0)
	>90	221	1	54.1 (33.7)
Age groups (years)	<50	18,941	52	48.8 (29.8)
	>50	17,525	48	59.7 (30.8)
Area	North Dublin	1758	5	50.8 (30.9)
	South Dublin	18,827	52	54.5 (31.7)
	West Dublin	8310	23	50.6 (30.3)
	Outside Dublin	7571	21	55.8 (29.5)
Season	Winter	8101	22	46.8 (30.3)
	Spring	10,321	28	48.9 (30.4)
	Summer	9353	26	58.6 (30.5)
	Autumn	8691	24	63.0 (29.4)
Gender	Male	10,335	28	49.7 (29.0)
	Female	261	72	55.1 (31.3)
Total		36,466		53.8 (30.78)

**Table 2 nutrients-12-02663-t002:** Area Demographics.

Area	N	GM Mean (SD)	<50 Years (%)	Female (%)
Dublin 1	290	45.5 (31.4)	88	73
Dublin 2	639	49.5 (29.6)	74	72
Dublin 3	219	54.9 (33.5)	78	80
Dublin 4	662	58.8 (33.0)	57	74
Dublin 5	111	54.6 (30.7)	75	77
Dublin 6/6W	5273	58.6 (31.1)	44	76
Dublin 7	496	51.4 (31.5)	80	74
Dublin 8	2915	49.1 (31.9)	61	74
Dublin 9	173	51.0 (29.5)	77	79
Dublin 10	846	50.8 (30.5)	42	72
Dublin 11	142	46.2 (27.3)	86	75
Dublin 12	3375	54.1 (31.8)	36	76
Dublin 13/17	107	52.0 (35.9)	82	78
Dublin 14/16	4129	59.9 (30.8)	36	70
Dublin 15	480	50.3 (31.5)	75	70
Dublin 18	372	54.0 (36.9)	72	66
Dublin 20	1353	54.3 (29.8)	34	64
Dublin 22	719	49.8 (29.5)	61	68
Dublin 24	750	52.3 (30.1)	58	73
North Co. Dublin	220	50.7 (27.7)	77	79
South Co. Dublin	616	55.9 (29.7)	58	72
Lucan, Co. Dublin	5008	46.5 (30.8)	64	69
North Co. Kildare	5734	56.1 (27.5)	49	68
Rest of Kildare	757	56.5 (29.0)	52	69
Co. Meath	502	56.2 (31.3)	66	75
Co. Wicklow	164	58.2 (28.8)	58	71
Rest of Leinster	209	54.1 (29.9)	67	72
Rest of Ireland	205	53.8 (30.5)	78	70
Total	36,466	53.8 (30.8)	52	72

Note: County abbreviated as Co.

**Table 3 nutrients-12-02663-t003:** 25(OH)D categorised by year, age and gender.

Year	N	Total (%)			Age							Gender						
					<50 Years (%)			>50 Years (%)			*p*-Value	Female (%)			Male (%)			*p*-Value
		<30	30–49	>125	<30	30–49	>125	<30	30–49	>125		<30	30–49	>125	<30	30–49	>125	
2014	5394	13	21	3	18	25	3	9	18	4	<0.001	12	20	3	16	25	2	<0.001
2015	6010	17	24	2	22	29	2	12	20	3	<0.001	16	23	3	19	29	1	<0.001
2016	7625	16	23	3	20	26	2	11	20	4	<0.001	15	21	4	18	27	2	<0.001
2017	8869	14	23	3	16	26	2	11	18	4	<0.001	12	22	3	17	25	2	<0.001
2018	8568	15	24	3	17	27	3	12	20	4	<0.001	14	23	4	17	28	2	<0.001
Total	36,466	15	23	3	18	27	2	11	19	4		14	22	3	17	27	2	

**Table 4 nutrients-12-02663-t004:** 25(OH)D status by area and season.

General Area	N	GM Mean (SD)	Winter			Spring			Summer			Autumn		
			*n*	<30%	30–49%	*n*	<30%	30–49%	*n*	<30%	30–49%	*n*	<30%	30–49%
Dublin 1	290	45.5 (31.4)	76	37	25	88	26	28	66	9	14	60	30	13
Dublin 2	639	49.5 (29.6)	157	34	20	170	21	29	151	9	25	161	19	22
Dublin 3	219	54.9 (33.5)	52	27	23	61	15	21	50	2	22	56	11	38
Dublin 4	662	58.8 (33.0)	132	19	23	181	21	23	169	7	13	180	10	15
Dublin 5	111	54.6 (30.7)	26	8	42	35	17	31	25	16	12	25	12	16
Dublin 6/6W	5273	58.6 (31.1)	1093	19	25	1488	17	22	1256	6	15	1436	8	17
Dublin 7	496	51.4 (31.5)	88	34	27	133	23	27	150	14	12	125	10	22
Dublin 8	2915	49.1 (31.9)	661	28	25	780	27	26	727	12	20	747	17	24
Dublin 9	173	51.0 (29.5)	49	24	29	41	22	32	41	0	22	42	14	21
Dublin 10	846	50.8 (30.5)	173	32	26	210	24	27	235	8	21	228	11	19
Dublin 11	142	46.2 (27.3)	33	39	39	39	18	41	41	12	20	29	17	17
Dublin 12	3375	54.1 (31.8)	734	22	24	1018	22	24	814	9	15	809	12	17
Dublin 13/17	107	52.0 (35.9)	26	23	27	34	24	35	23	4	22	24	17	13
Dublin 14/16	4129	59.9 (30.8)	924	18	23	1181	13	26	963	6	12	1061	8	16
Dublin 15	480	50.3 (31.5)	111	23	28	129	26	29	126	13	20	114	17	17
Dublin 18	372	54.0 (36.9)	84	21	30	106	25	27	75	7	19	107	6	21
Dublin 20	1353	54.3 (29.8)	335	24	25	384	15	28	319	7	20	315	13	21
Dublin 22	719	49.8 (29.5)	129	28	32	218	24	26	202	11	19	170	18	25
Dublin 24	750	52.3 (30.1)	157	25	30	210	24	24	190	10	12	193	12	21
North Co. Dublin	220	50.7 (27.7)	44	36	39	66	18	23	64	5	25	46	13	20
South Co. Dublin	616	55.9 (29.7)	133	23	26	175	13	29	158	6	17	150	12	18
Lucan, Co. Dublin	5008	46.5 (30.8)	1112	32	27	1353	28	28	1204	15	20	1339	18	22
North Co. Kildare	5734	56.1 (27.5)	1329	18	30	1712	18	26	1249	4	15	1444	6	20
Rest of Kildare	757	56.5 (29.0)	170	23	27	215	14	27	166	4	16	206	9	19
Co. Meath	502	56.2 (31.3)	132	22	25	141	19	23	104	5	14	125	6	25
Co. Wicklow	164	58.2 (28.8)	36	33	19	36	11	19	40	0	8	52	10	21
Rest of Leinster	209	54.1 (29.9)	52	27	29	66	17	36	49	6	14	42	2	19
Rest of Ireland	205	53.8 (30.5)	53	25	28	51	16	29	34	9	18	67	10	24
Total	36,466	53.8 (30.8)	8101	23	26	10,321	20	26	8691	8	16	9353	11	20

**Table 5 nutrients-12-02663-t005:** 25(OH)D Status by location, age and gender.

	Age							Gender						
	<50 Years			>50 Years				Female			Male			
Vitamin D (%)	<30	30–49	>125	<30	30–49	>125	*p*-Value	<30	30–49	>125	<30	30–49	>125	*p*-Value
Dublin 1	26	21	2	17	28	6	0.420	22	22	3	32	23	1	0.547
Dublin 2	17	27	3	22	25	2	0.140	17	28	3	21	23	2	0.611
Dublin 3	14	30	6	4	21	6	0.025 *	11	23	7	14	45	2	0.009 *
Dublin 4	14	21	3	12	16	6	0.036 *	12	17	5	16	25	3	0.004 *
Dublin 5	11	29	4	18	21	0	0.046 *	9	24	4	23	38	0	0.555
Dublin 6/6W	15	24	3	9	18	4	<0.001 *	11	19	5	15	26	2	<0.001 *
Dublin 7	18	26	4	12	13	4	0.672	16	23	5	19	24	2	0.001 *
Dublin 8	23	27	2	14	21	4	0.002 *	19	24	3	24	27	2	<0.001 *
Dublin 9	14	26	5	10	35	0	0.466	12	26	4	17	36	0	0.549
Dublin 10	18	24	3	16	24	2	0.291	16	23	3	20	26	2	0.349
Dublin 11	19	34	2	15	20	5	0.994	18	33	2	19	31	3	0.211
Dublin 12	18	26	2	14	18	3	<0.001 *	15	20	3	18	26	1	<0.001 *
Dublin 13/17	16	26	6	26	21	0	0.962	18	27	5	17	21	4	0.645
Dublin 14/16	13	26	2	9	17	5	<0.001 *	10	18	5	12	25	1	<0.001 *
Dublin 15	20	29	3	10	16	3	0.656	17	25	4	18	28	1	0.001 *
Dublin 18	16	25	3	10	23	6	0.068	14	20	4	16	32	4	0.042 *
Dublin 20	16	28	2	13	23	3	<0.001 *	14	23	4	14	28	1	<0.001 *
Dublin 22	23	28	2	13	23	4	0.005 *	18	22	3	20	34	2	<0.001 *
Dublin 24	22	26	2	9	17	2	0.004 *	15	21	2	21	27	3	<0.001 *
North Co. Dublin	17	26	2	10	32	0	0.215	17	26	1	9	30	4	0.214
South Co. Dublin	13	27	1	10	19	4	0.254	10	25	3	17	22	2	0.021 *
Lucan, Co. Dublin	27	28	2	12	21	3	<0.001 *	21	24	2	25	28	1	<0.001 *
North Co. Kildare	14	28	2	8	20	3	<0.001 *	10	23	2	12	26	2	<0.001 *
Rest of Kildare	16	25	3	8	21	3	0.303	12	22	3	11	26	2	<0.001 *
Co. Meath	14	27	3	8	17	3	0.328	11	24	3	17	22	2	0.009 *
Co. Wicklow	16	18	2	6	19	3	0.329	9	20	2	19	15	4	0.350
Rest of Leinster	16	28	1	7	25	4	0.521	13	25	3	14	32	0	0.167
Rest of Ireland	14	30	4	9	20	4	0.052	10	28	5	20	28	3	0.425
Total	18	27	2	11	19	4		14	22	3	17	27	2	

Notes: Vitamin D categories expressed as % for <30 nmol/L, 30–49 nmol/L, >125 nmol/L.(*) Indicates significance ***p*** < 0.05 level. County abbreviated as Co.

**Table 6 nutrients-12-02663-t006:** Predictors of vitamin D deficiency <30 nmol/L) and insufficiency (30–49 nmol/L) in multinomial regression.

Deficient vs. Sufficient	*n*	B (SE)	OR	Lower	Upper	Effect (%)	*p*-Value
Intercept		1.224 (0.072)					
Age	36,466	0.023 (0.001)	0.977	0.976	0.979	−2	<0.001 *
Female	26,161	0.39 (0.033)	0.677	0.634	0.723	−32	<0.001 *
Winter	8101	−1.49 (0.049)	4.435	4.030	4.881	344	<0.001 *
Spring	10,321	−1.275 (0.048)	3.58	3.261	3.930	258	<0.001 *
Autumn	8691	−0.418 (0.052)	1.519	1.372	1.681	52	<0.001 *
North Dublin	1758	−0.431 (0.076)	1.539	1.327	1.786	54	<0.001 *
South Dublin	18,827	−0.35 (0.043)	1.419	1.304	1.543	42	<0.001 *
West Dublin	8310	−0.776 (0.047)	2.172	1.981	2.382	117	<0.001 *
Outside Dublin	7571						
Insufficient vs. Sufficient							
Intercept		0.479 (0.058)					
Age	36,466	0.017 (0.001)	0.983	0.982	0.985	−2	<0.001 *
Female	26,131	0.387 (0.029)	0.679	0.642	0.719	−32	<0.001 *
Winter	8101	−0.888 (0.04)	2.430	2.247	2.628	143	<0.001 *
Spring	10,321	−0.807 (0.038)	2.241	2.081	2.414	124	<0.001 *
Autumn	8691	−0.264 (0.04)	1.303	1.205	1.408	30	<0.001 *
North Dublin	1758	−0.085 (0.066)	1.089	0.957	1.239	9	<0.001 *
South Dublin	18827	−0.019 (0.034)	1.019	0.953	1.089	2	0.194
West Dublin	8310	−0.256 (0.039)	1.291	1.195	1.395	29	0.586
Outside Dublin	7571						

Notes: (*) Indicates significance at <0.05 level.

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
