# Peer review of "Geomapping Vitamin D Status in a Large City and Surrounding Population—Exploring the Impact of Location and Demographics"

_nutrients, 2020, doi:10.3390/nu12092663_

Round 1

Reviewer 1 Report

This most interesting paper retrospectively evaluates requested 25OHD analyses from GP's over 5 years restricted to a narrow latitudinal expanse of North Ireland (Dublin). The authors show the surprising impact of seasonality, gender, socioeconomic and most surprising age effect on vitamin D status in populations located very close to one another. A number of strengths in this study enable comparisons to other areas of Ireland and possibly elsewhere, the most important of which is the use of the validated, standardized (gold standard) analytical method for 25OHD (LC-MS/MS with NIST internal standard and DEQAS accreditation) and the use of the IOM reference cut-off values for defining deficiency and insufficiency which limits variability in these measures over time and locations.

The use of geo-mapping is unique and original which enabled teasing out the finding of greatest deficiency/insufficiency in the oldest and youngest tested, those most likely to not take supplements or have impaired endogenous synthesis due to work indoors, frailty, etc. This gives me pause to consider that the averaging of vitamin D measures over large geographic population areas may be giving us very false ideas about the extent of vitamin D deficiency in some specific pockets, regions, states/provinces, or countries, especially large ones like the USA. I do not know if it would be possible to geo-map these data according to race/ethnicity; but, this would certainly be valuable information in targeting suitable/acceptable food vehicles for vitamin D fortification. Mandatory vitamin D food fortification may not be suitable or acceptable if limited to one food category, such as milk. Dietary intake information is also needed to determine what foods are consumed.

Author Response

Thank you for reviewing this manuscript and for providing your valuable feedback. 

The authors are in agreement that averaging of vitamin D values over large populations gives a false impression of vitamin D deficiency and that geomapping may be a useful tool to further clarify its prevalence in larger populations such as the USA.

The authors also agree that race/ethnicity information is an important consideration for the targeting of public health solutions. 

While ethnicity was not available in this dataset due to its anonymised nature, a subsequent qualitative study is underway in which questionnaire data will be collected to assess risk of Vitamin D deficiency due to contributory factors such as socioeconomic status, dietary intakes and ethnicity.

Reviewer 2 Report

In this interesting study the authors aimed to describe the Vitamin D status in a large population from Dublin Area. The study is well designed and the analysis well performed and it is the largest study in Europe to geomap vitamin D.
Overall the paper is well designed and written, I have some advice to improve this valuable work:

At the paragraph 4.3, line 339, when you write about potential co-morbidities related to vitamin D deficiency, it will be interesting to mention the interaction between renin–angiotensin–aldosterone system (RAAS) and vitamin D and to mention the function of Vitamin D Receptors (VDRs). I suggest you the RAAS paragraph of the work Gembillo G et al Role of Vitamin D Status in Diabetic Patients with Renal Disease. Medicina (Kaunas). 2019 Jun 13;55(6). pii: E273. doi: 10.3390/medicina55060273. Review. PubMed PMID: 31200589; PubMed Central PMCID: PMC6630278. In this work has been described the interaction of VDR with RAAS, “The interaction of plasma renin and vitamin D is tightly connected with the VDR status: in case of vitamin D deficiency there is a reduced transcription of VDR and an enhanced degradation of unliganded VDR, with a decrease in both unliganded and liganded VDR. This deficiency of liganded VDR, as previously mentioned, would improve the transcription of renin whereas the lack of unliganded VDR would enhance the transcription of angiotensinogen and Angiotensin II Type I Receptors (AT1Rs) via modulation of p53 expression”. I also suggest you the paper of Santoro et al. “Interplay of vitamin D, erythropoiesis, and the renin-angiotensin system.” BioMed research international vol. 2015 (2015): 145828. doi:10.1155/2015/145828”.
Another aspect that can be emphatized is the role of vitamin D as inflammation mediator, by the Megalin-Cubilin-Amnionless and FGF23-Klotho axis, you can find more information in the article “Gembillo G et al Protective Role of Vitamin D in Renal Tubulopathies. Metabolites. 2020 Mar19;10(3). pii: E115. doi: 10.3390/metabo10030115. Review. PubMed PMID: 32204545; PubMed Central PMCID: PMC7142711.”

Minor revisions:
Line 49: “A significant proportion of the population “are” at risk of deficiency” […]. The plural verb “are” does not appear to agree with the singular subject proportion. Consider changing the verb form for subject-verb agreement.
Line 73: “community dwelling” is missing a hyphen. Consider adding it.
Line 107: “chromatography tandem” is missing a hyphen.
Line 370: “cross sectional” is missing a hyphen.

Author Response

Thank you for reviewing this manuscript and for providing your valuable feedback. 

Thank you for the suggested references which the authors have considered for inclusion. 

The following line and reference (Gembillo G et al Role of Vitamin D Status in Diabetic Patients with Renal Disease. Medicina (Kaunas). 2019 Jun 13;55(6). pii: E273. doi: 10.3390/medicina55060273. Review. PubMed PMID: 31200589; PubMed Central PMCID: PMC6630278 ) has been added

(added line 344) Furthermore Vitamin D is a known regulator of cardiovascular and renal function mediated via the interaction of Vitamin D receptors within the renin–angiotensin–aldosterone system (RAAS), highlighting the multisystemic effects of vitamin D deficiency.  Gembillo G et al (2019)

The following corrections have been made for the highlighted typos:

Line 49: “A significant proportion of the population “are” at risk of deficiency” has been changed to “A significant proportion of the population is at risk of deficiency.

Line 73: “community dwelling” has now been hyphenated
Line 107: “chromatography tandem” has now been hyphenated

Line 370: “cross sectional” has now been hyphenated